# Basic Motor Competencies of (Pre)School Children: The Role of Social Integration and Health-Related Quality of Life

**DOI:** 10.3390/ijerph192114537

**Published:** 2022-11-05

**Authors:** Kathrin Bretz, Harald Seelig, Ilaria Ferrari, Roger Keller, Jürgen Kühnis, Simone Storni, Christian Herrmann

**Affiliations:** 1Physical Education Research Group, Zurich University of Teacher Education, 8090 Zurich, Switzerland; 2Department of Sport, Exercise and Health, University of Basel, 4052 Basel, Switzerland; 3Centre for Inclusion and Health in Schools, Zurich University of Teacher Education, 8090 Zurich, Switzerland; 4Expert Group Physical Education, Schwyz University of Teacher Education, 6410 Goldau, Switzerland; 5Didactics of Physical Education, University of Applied Sciences and Arts of Southern Switzerland, 6600 Locarno, Switzerland

**Keywords:** kindergarten, sport, health, motor skills, physical education, well-being

## Abstract

In (pre)school, children acquire and deepen their basic motor competencies (BMCs) and interact with peers and friends. BMCs are a central developmental goal in childhood and the prerequisite for participation in sportive aspects of social life. Both motor competencies and social integration are linked to children’s health-related quality of life (HRQoL). The aim of the present study was to describe the connection between BMCs, social relationships, and aspects of HRQoL in (pre)school children. In this study, the BMCs of N = 1163 preschool children (*M* = 5.7 years, *SD* = 0.57, 52% boys) and N = 880 first and second graders (*M* = 7.5 years, *SD* = 0.58, 51% boys) were tested. The children’s social integration was assessed by the teachers; the HRQoL was recorded from the parents’ perspective. In both preschool and primary school, children with better BMCs also showed higher values in their social integration. Moreover, the results indicated a connection between BMCs and general HRQoL in primary school and BMCs and physical well-being in preschool. As BMCs, social integration, and HRQoL seem to be connected in (pre)school, this should be considered both from developmental and health-oriented perspectives, as well as for physical education (PE) lessons.

## 1. Introduction

Throughout childhood, children develop and extend their basic motor repertoire in contexts of social interaction (e.g., (pre)school, interactions with peers, sport clubs). Basic motor competencies (BMCs) are necessary for participation in the culture of sport and exercise [1,2]. They facilitate a basic capacity for the development of higher competency levels and further participation in sports and exercise [2]. BMCs are also the prerequisite for acquisition of sport-specific motor skills and positively influence a physically active lifestyle over an individual’s lifespan [2]. BMCs further lead to the development of a large repertoire of movement skills [3]. This process is strongly influenced by opportunities for practice [4]. Preschools and primary schools should offer situations in which children can extend their BMCs [5]. Physical education (PE) classes with various movement situations are particularly suitable, as all children participate in them in contrast to extracurricular activities. The development of BMCs is a core development task in preschool and primary school and is also addressed in PE curricula [5].

For all children, school is an expanded social environment with new tasks and challenges [6]. Interpersonal relationship skills are addressed in the curricula and defined as a central life skill by the World Health Organization (WHO) [6,7]. Interpersonal relationship skills help children to relate to the people around them in a positive way. Moreover, they are acquired in contexts involving social interaction and are important for making friends and getting involved in peer groups [6,8]. Social interactions and peer relationships are central to healthy child development [9,10]. Friendships are associated with a variety of positive psychological and behavioral outcomes for children, which continue into early adulthood [11]. Children choose friends and peers by participating in similar activities (e.g., sports, music, art) or interacting with those who behave similarly [12]. In early childhood, children are more likely to choose same-gender friendships and different genders show different play behavior, which can already be observed at preschool age [13]. While boys more often engage in physically active games, girls desire friendly closeness and cohesion more than boys [13]. Girls exert a lot of effort establishing and maintaining positive social relationships and spend time in smaller groups with closer friends [12,14].

In order to ensure adequate learning opportunities for every child, it is important to promote social integration in the class in addition to subject-specific competencies in PE [15]. This is particularly important in the context of sport and play situations, as this can only happen in the interplay between BMCs and social integration [9].

Positive associations between BMCs and social relationships have been found in preschool-aged children [16]. In addition, 9 to 12 year old children with poorer motor competencies have been found to be less preferred by their peers in both play and classroom settings [17]. Children with developmental coordination disorder, in particular, have been found to be less socially integrated and more likely to experience exclusion in class [18].

Integration with peers and interactions with friends are highly important for children’s quality of life, as well as their well-being [19]. Health-related quality of life (HRQoL) is a multidimensional and complex construct described as an individual’s perception of his or her position in life [20]. It includes physical, emotional, mental, social, and behavioral components of well-being and functioning from the subjective perspective [20]. The assessment of HRQoL in children has increasingly become the focus of health research [21]. An assessment of HRQoL can be used to identify subgroups or individuals who are at higher risk of health problems [22]. Therefore, HRQoL is especially examined in children with special needs or diseases, such as developmental coordination disorder or chronic illness [23,24].

Both BMCs and social integration have a positive influence on children’s mental and physical health [25,26]. Studies investigating the determinants of HRQoL in (pre)school children show that children with higher motor competencies have better HRQoL levels [24,26,27,28]. Moreover, children with low motor competencies show a higher risk of negative interpersonal (peer problems) and intrapersonal (low self-assessment) consequences at the psychosocial level, which can result in worse mental health and well-being [18,29,30]. Integration with peers and interactions with friends are highly important for quality of life, as positive relationships with friends have a strong effect on children’s subjective well-being [25,28].

However, little research has been conducted on the connections between (basic) motor competencies, social integration, and HRQoL in children, especially in preschoolers [26,31]. As previous studies have investigated the relationship between motor competencies and HRQoL mainly in children with DCD or special needs, there is a need to investigate this relationship in normally developing children as well. What should be taken into account in particular is the idea of participation, especially in PE. The aim of this study was to investigate the relationship between BMCs, social integration, and HRQoL in children in their first years of (pre)school.

## 2. Materials and Methods

The present study was a cross-sectional study based on the first measurement point for a longitudinal research project, funded by the Zurich University of Teacher Education and Health Promotion Switzerland (Gesundheitsförderung Schweiz, GFCH) and utilized convenience sampling. Although it was not a representative sample for Switzerland, we ensured that all three language regions, as well as urban and rural areas, were equally represented in the sample. In spring and summer 2021, we measured the BMCs of preschoolers (4–6 years) and children from the first and second grades (6–8 years) in the German-, Italian-, and French-speaking parts of Switzerland. Preschool in Switzerland is part of mandatory schooling and includes a two-year entrance stage for primary school.

### 2.1. Participants

In total, we contacted the parents or legal guardians of 1840 preschoolers and 1163 children from the first two years of primary school in the German-, Italian-, and French-speaking parts of Switzerland. For preschoolers, 1334 parents (72.5%) gave their written consent for their children to participate in the study and sent back the questionnaire. Inclusion criteria were the presence of consent for the assessment of the BMC test and the parent questionnaire. Age ranges were formed based on the dates of entry to preschool (55–80 months) and primary school (77–105 months) in order to exclude much younger and older children from the study. We were thus able to include 1163 preschoolers (M = 5.7 years, SD = 0.57, 52% boys) from 95 classes (average class size, *n* = 13). In the first and second grades of primary school, 901 parents (77.5%) agreed to their children’s participation in the study. We included 880 (M = 7.5 years, SD = 0.58, 51% boys) children from the first and second grades from 64 classes in the study (average class size, *n* = 14). We received assessments from the teachers (M = 39.7 years, SD = 10.2, 90% female teachers) and the parents (M = 38.5 years, SD = 5.9, 76% female).

### 2.2. Test Instruments and Data Collection

Basic motor competencies (BMCs; children tests):

To measure BMCs, we used the MOBAK test instruments for preschool (MOBAK-KG) and the first two years of primary school (MOBAK-1-2). The MOBAK instrument is a curriculum-valid instrument that measures the level of BMC and can be used easily in PE lessons [1,32]. Moreover, it is oriented toward the elementary learning goals of PE (e.g., [5]). The BMCs in the two competence areas of self-movement and object movement (Table 1; for details, see [1,32]) are measured via four items each. A standardized task with corresponding evaluation criteria is described per item. The children performed two trials per test item (six trials for the throwing and catching items). Both attempts were rated dichotomously (0 = fail, 1 = successful). The individual results per test item were summed up to calculate the final item score (0 points = no successful attempts, 1 point = one successful attempt, 2 points = two successful attempts). The throwing and catching scores were calculated differently. In these cases, 0–2 successful attempts were scored as 0 points, 3–4 successful attempts as 1 point, and 5–6 successful attempts as 2 points. For each competency domain, a maximum sum score of eight points could be achieved (for details, see [1,32]). The data collection took 30–40 min and was carried out during a regular PE lesson of 45 min duration. The classes were split up and an examiner led three to four children through the eight test stations and gave a standardized explanation and one demonstration of each test item.

The factorial validity of the MOBAK instruments for preschool and primary education has already been investigated and confirmed in various studies (e.g., [33,34]).

Social integration (PIQ; teacher questionnaires):

The teachers measured the children’s social integration using the subscale of the perception of inclusion (PIQ) questionnaire [15]. The teachers rated the children individually via four items (e.g., “He/she gets along very well with his/her classmates.”) on a four-point scale. The teachers received the questionnaire for each child in advance along with the information on the study. We asked the teachers to complete the questionnaire and bring it with them on the day of the MOBAK test. The Cronbach’s alpha of the scale was calculated for preschool (0.82) and primary school (0.83) and showed satisfactory internal consistency [35]. The factorial validity of the instrument was confirmed in a validation study by Venetz and colleagues. [15].

General health-related quality of life (general HRQoL; parent questionnaires):

The low reading literacy of children, especially in early childhood, has led to the development of instruments that measure children’s HRQoL via parental assessments [36]. General health-related quality of life (HRQoL) was measured via the KIDSCREEN-10 instrument [36,37] in a subsample of N = 943 preschool children and the total sample of N = 880 primary school children (subsample 1), with a short version used in one canton due to the construction of the questionnaire. This instrument contains ten items (e.g., “Has your child felt sad?”) and provides a valid measure of a general HRQoL factor. Moreover, the parents filled out the children’s date of birth and gender. The parents received the questionnaire along with the declaration of consent, both of which were collected by the teachers. The internal consistency of the KIDSCREEN-10 instrument was acceptable, with a Cronbach’s alpha of 0.73 for preschool and 0.76 for primary school (overall 0.74) [35]. For the analyses, the sum score (10–50) was transformed into the t-value (mean: 50, standard deviation: 10). Higher values indicate a higher general HRQoL [36,37].

Physical well-being (parent questionnaires):

In a subsample of N = 348 preschool children (subsample 2), the physical well-being subscale of the KIDSCREEN-27 instrument [36] was used exploratively. The subscale consists of five items (e.g., “Has your child felt fit and well?”) and had an acceptable Cronbach’s alpha of 0.71. For the analyses, the sum score (5–23) was transformed into the t-value (mean: 50, standard deviation: 10). Higher values indicate higher physical well-being [37].

### 2.3. Data Analysis

SPSS 28 was employed for the data editing, descriptive statistics, t-tests, and Cronbach’s alpha estimations [38]. Descriptive statistics were calculated for all variables. T-tests were used to calculate differences between boys and girls in the variables of interest. In addition to the 95% confidence intervals, Cohen’s d was calculated to examine the strength of the differences. Therefore, effect sizes were interpreted following Cohen (1988) as small (d = 0.10), medium (d = 0.50), and large (d = 0.80) [39]. We used Mplus 8.4 to perform multivariate analyses [40]. We calculated interclass correlations (ICCs) to test the influences of the multilevel structure (pupils from different classes) due to class associations. A high ICC value means that there are large differences between classes for the corresponding characteristics, the cause of which is to be sought at the class level (e.g., class composition). Raudenbush and Bryk (2002) recommend accounting for the multi-level structure of the data for advanced analyses with ICCs > 0.05 [41].

*Model 1:* In this first model, we used structural equation models to examine the relationships between the two MOBAK factors self-movement and object movement, social integration, and general HRQoL, with age as a covariate. Self-movement and object movement, as well as social integration, were included as latent factors.

Following Ravens-Sieberer and colleagues, we summed up general HRQoL, transformed it into the t-value, and included it as a manifest variable in the model [37] (Figure 1). This model was separately examined for both age groups of interest (MOBAK-KG, model 1a; MOBAK-1-2, model 1b). Since KIDSCREEN-10 was not used (in its entirety) at all study locations, model 1 was calculated for a subsample of N = 943 preschool children and N = 880 primary school children (subsample 1).

*Model 2:* Next, we re-calculated model 1 as a multigroup model to investigate the correlations between the model components separately for boys and girls. This allowed for a model test for boys and girls. All parameters were estimated freely. Only the factor structure was kept equal between boys and girls [42,43,44]. This served to ensure that the factor structure (numbers and types of latent factors and loadings) was the same for boys and girls. We calculated model 2 separately for both MOBAK-KG (model 2a) and MOBAK-1-2 (model 2b).

*Model 3:* In a subsample of N = 384 preschool children (subsample 2), we used the physical well-being subscale of the KIDSCREEN-27 instrument (5 items [36]) to assess the children’s physical well-being from the parents’ perspective. The sum score of the five items was t-transformed into a manifest variable. We used structural equation models to calculate the relationship between the latent factors self-movement, object movement, and social integration and the manifest variable physical well-being. Age was included as a covariate (Figure 2).

*Model 4:* We then re-calculated model 3 as a multigroup model for boys and girls. We examined the configural invariance in a multiple group model. This allowed for a model test for boys and girls simultaneously.

In all models, we treated the MOBAK test items as ordinal-scaled and the questionnaire items as interval-scaled data. Accordingly, we applied the mean- and variance-adjusted weighted least squares (WLSMV) estimator.

The “type = complex” function for nested datasets implemented in Mplus was needed to correct the standard error and ensure that dependencies within the multilevel structure (0.01 ≤ ICC ≤ 0.19; Table 2) were accounted for in all model estimations [41]. The goodness of fit of the models was assessed using fit indices proposed in the literature [45]. Effect sizes were interpreted as small (r > 0.10, β > 0.05), medium (r > 0.30, β > 0.25), and large (r > 0.50, β > 0.45) [39,46].

We accounted for missing values by generating model estimates using the full information maximum likelihood (FIML) procedure. This procedure prevents bias in the sample composition by preventing a reduction in the sample size [47].

## 3. Results

As the descriptive analyses (Table 2) show, there were already gender-specific differences in motor performance levels. Girls were better in self-movement, while boys performed better in object movement. Girls were rated as more socially integrated by their teachers in both preschool and primary school. In preschool, there were no gender differences regarding general HRQoL, whereas boys showed higher physical well-being than girls. In primary school, general HRQoL was higher in girls than in boys. The ICC values for BMCs, general HRQoL, and physical well-being were low (ICC < 0.05). The ICC value for social integration was 0.19. This means that there were large differences between the classes due to the assessment of social integration by the teacher at the class level.

### Latent Structural Equation Models

*Model 1*: Both model 1a (preschool) and model 1b (primary school) fit the data well (Table 3).

Table 4 shows that the associations of the latent constructs of BMC and social integration with general HRQoL differed between preschool and primary school children. A positive small to moderate association between children’s BMCs and their social integration (assessed by the teachers) was found in both preschool and primary school. In preschool, no correlations were found between general HRQoL (assessed by the parents) and BMCs or between general HRQoL and social integration.

In primary school, positive significant associations were found between general HRQoL and self-movement (*r* = 0.14, *p* = 0.005), as well as general HRQoL and social integration (*r* = 0.13, *p* < 0.001), whereas no correlations were found between general HRQoL and object movement. The results show that children who were better with self-movement and children who were better socially integrated obtained higher values for general HRQoL. The correlations with age show that older children had better BMCs and seemed to be better socially integrated than younger children. No correlation with age was found for general HRQoL.

*Model 2:* Taking model 1 as a starting point, the correlations between the latent factors, as well as with age as a covariate, were calculated for both genders separately in a multigroup model (Table 5). This model achieved a good fit for preschool (model 2a, Table 3) and primary school (model 2b, Table 3). Deviation estimates for boys and girls appeared to be comparable in preschool (boys: χ^2^ = 106.924, *n* = 484; girls: χ^2^ = 94.146, *n* = 459) and within primary school (boys: χ^2^ = 62.191, *n* = 450; girls: χ^2^ = 117.932, *n* = 430).

In model 2, the results were similar for boys and girls. Boys and girls with better BMCs were rated higher for their social integration by their teachers in both preschool and primary school. Regarding general HRQoL, there were no associations with BMCs or social integration for either boys or girls in preschool. For primary school, significant relationships with general HRQoL, as assessed by the parents, were only found for self-movement among girls (*r* = 0.18, *p* = 0.007) and social integration among boys (*r* = 0.16, *p* = 0.004). The finding that age was positively associated with BMCs and social integration was evident for girls and boys.

*Model 3:* The structural equation model with object movement, self-movement, social integration, and the subscale physical well-being (t-value of subscale sum score) achieved a good model fit (Table 3). A high correlation between object movement and self-movement was found (*r* = 0.79, *p* < 0.001). As already became clear from model 1 and model 2, older children showed higher BMCs and were assessed as being better socially integrated. Moreover social integration was significantly correlated with object movement (*r* = 0.29, *p* < 0.001) and self-movement (*r* = 0.40, *p* < 0.001). No significant correlation was found between social integration and physical well-being. The BMCs of the children were positively correlated with both object movement (*r* = 0.20, *p* = 0.004) and self-movement (*r* = 0.29, *p* < 0.001) (Table 6).

*Model 4:* Taking model 3 as a starting point, the correlations between the factors, as well as with age as a covariate, were calculated for both genders separately. This multigroup model achieved a good fit (Table 3). Separate deviation estimates were χ^2^ = 67.209, *n* = 198 for boys and χ^2^ = 108.436, *n* = 186 for girls. For both genders, the children’s social integration, as assessed by their teachers, was moderately related to BMCs. Correlations with BMCs could also be found for physical well-being. Both boys and girls showed high correlations between self-movement and physical well-being (*r* = 0.35/0.34, *p* < 0.001). In the competency domain of object movement, a significant correlation was only observed for boys (*r* = 0.21, *p* = 0.009). No significant correlation was found between physical well-being and social integration (Table 7).

## 4. Discussion

The objective of the present study was to investigate the relationship between BMCs, social integration, and health-related quality of life in (pre)school children. The results of this study support earlier findings that children with poor BMCs are less integrated socially and show poorer general HRQoL in primary school and physical well-being in preschool.

Moreover, girls’ general HRQoL was rated higher than boys’ general HRQoL in primary school, whereas there were no gender differences in preschool. Physical well-being, on the other hand, was rated higher for boys than for girls. Differences in HRQoL between age and gender have only previously been studied from eight years onwards [48].

The findings of this study are consistent with previous studies indicating that children with lower motor competencies show lower HRQoL and are less integrated socially [16,26]. The relationship between motor competence and HRQoL has mostly been investigated in children with developmental coordination disorder, as these children are more likely to have psychological issues, which may result from poor social skills or decreased quality of life [18,24,49,50]. Moreover, children with DCD show lower scores in HRQoL than typically developing children [24]. Redondo-Tébar and colleagues (2021) studied the relationship between motor competence and HRQoL in a sample of typically developing children and found a positive association between HRQoL and motor competence [26]. In contrast to previous studies in which motor competence instruments were used in a clinical context (e.g., MABC-2 [24]), our study used curriculum-valid instruments that examine BMCs in self-movement and object movement.

Children with better BMCs seemed to be better integrated in both preschool and primary school, although this correlation was higher in preschool. This could have been due to the fact that activities other than play and sports become more important for friendships in primary school. From primary school onwards, extracurricular activities, such as musical or artistic activities, are increasingly offered, and these activities could become more important for friendships with increasing age. The increasing importance of academic achievement in school could also be a reason for the lower correlations.

Differences in the association between BMCs and general HRQoL were found between preschool and primary school. Whereas no correlations between BMCs and general HRQoL were found in preschool, primary school children with better performance in self-movement also showed higher values in general HRQoL. No connection with object movement was found. It is possible that general HRQoL in preschool is more strongly influenced by other factors, such as family factors (e.g., parents, siblings). Moreover, it could be that BMC is related to general HRQoL in more informal play settings (e.g., outside, with friends or siblings).

Due to the fact that motor competencies may be important for children’s physical well-being, we additionally used the physical well-being subscale of KIDSCREEN-27 in a subsample of N = 384 preschool children [26,36]. Physical well-being was higher in children with better BMCs, with a stronger association for self-movement than for object movement. The results indicate a significant relationship between BMCs and physical well-being, which has already been demonstrated in other studies [26,50].

Social integration and interaction with friends and peers are important factors for children’s well-being, since popularity, mutual friendships, and engagement in social play are positively associated with children’s quality of life [19,28]. In the present study, primary school children who were assessed to be better integrated socially also showed higher values in general HRQoL, although the association was stronger in boys. In accordance with previous studies, it appears that children who seem to be better socially integrated show higher general HRQoL.

One strength of the study was that the investigated constructs (BMCs (motor competence test [1,32]), social integration (teacher perspective [15]), and general HRQoL or physical well-being (parent perspective [36,51])) were examined from different perspectives. Thus, we took into account the perspective of the child but also those of the parents and teachers, as home and school are important environments in children’s everyday lives. Another strength was the high sample size achieved in this study. Nevertheless, a few limitations should be pointed out. While the KIDSCREEN-10 instrument is a valid measurement tool for general HRQoL and is especially useful in identifying subgroups of children who are at risk for health problems, it does not represent most of the dimensions captured in KIDSCREEN-27 [51]. This suggests that the different dimensions of the multidimensional construct of health-related quality of life should be considered in further studies. As was evident in the subsample, physical well-being is related to both BMCs and social integration. It should also be taken into account that the KIDSCREEN instrument is a validated instrument for children above eight years, and a validation study in a younger cohort has yet to be conducted [51]. Moreover, we used the teacher and parent perspective to assess social integration, general HRQoL, and physical well-being because of the young age of the children. Factors that influence HRQoL, such as socioeconomic status [52], should also be considered in future studies.

Due to the cross-sectional study design, it was not possible to identify the direction of causality. Accordingly, future longitudinal studies should examine the extent to which (basic) motor competencies influence children’s social integration and HRQoL and vice versa and how (basic) motor competencies can be targeted.

The findings of the study raise the pedagogical–didactic question of how to design a careful and effective setting for PE in (pre)school. As early as preschool, there are differences between boys and girls regarding BMCs that cannot be attributed to gender alone [53,54]. Teachers should be aware of gender-specific sports socialization and try to remove gender-specific role models (boys play with the ball, girls do gymnastics) in PE. This can happen, for example, through a polysportive approach or the inclusion of different combinations of movement, balls, or equipment in PE.

As PE addresses both BMCs and interdisciplinary competencies, such as interpersonal relationship skills, it should be held in an inclusive setting that promotes not only learning outcomes but also interpersonal relationship skills. Both BMCs and interpersonal relationship skills seem to be important for children’s social integration. For this purpose, PE classes should promote BMCs in social situations (e.g., open learning tasks that can be solved in a group and that do not reward the individual’s performance), as well as interaction with classmates. This could involve the creation of learning tasks for children with different levels of BMCs in which children can vary the difficulty, find different ways of solving the tasks, and cooperate with other children. This could help children to integrate better and feel more comfortable in the class setting. Teachers and practitioners should be aware of the connections between BMCs, social integration, and HRQoL in children. Children with poor motor competencies in particular should be encouraged to participate in sport activities and play interaction to improve both social integration and BMCs.

## 5. Conclusions

This study showed a positive association between BMCs, social integration, and HRQoL. In both preschool and primary school, children with better BMCs seemed to be better integrated socially in the class. Children with better BMCs showed better general HRQoL in primary school and better physical well-being in preschool. The improvement of BMCs and social integration in the class could contribute to higher HRQoL. Furthermore, additional opportunities for movement with an integrative character and opportunities for co-determination could be implemented both in PE and in extracurricular activities. Since both BMCs and social integration are linked to better HRQoL, we recommend creating inclusive situations in physical settings in which it is possible to improve both BMCs and social integration. Consequently, BMCs should be considered to improve both social integration and HRQoL.

## Figures and Tables

**Figure 1 ijerph-19-14537-f001:**
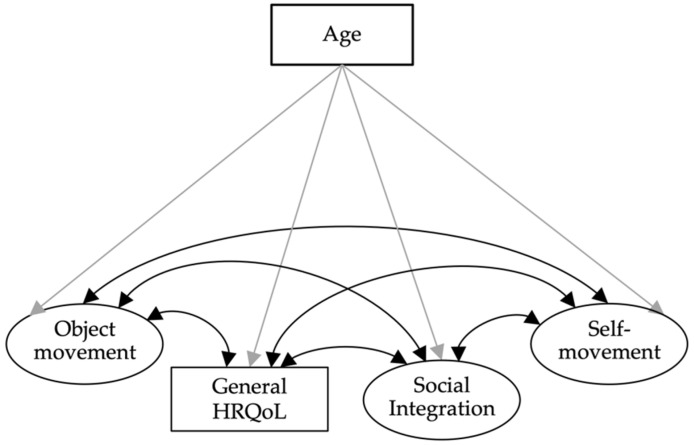
Model 1. Structural equation model with object movement, self-movement, general HRQoL, and social integration with the covariate age.

**Figure 2 ijerph-19-14537-f002:**
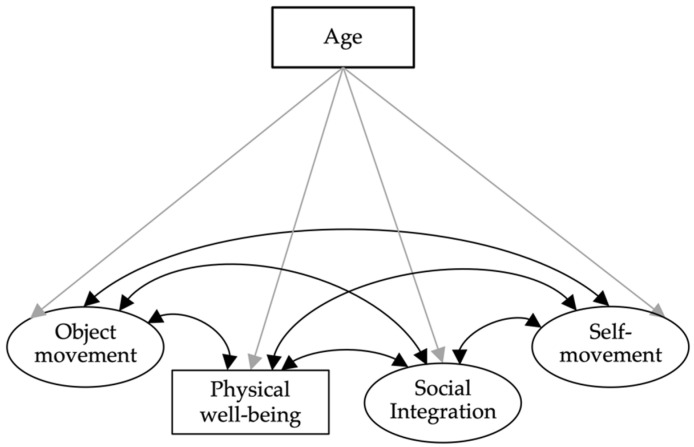
Model 3. Structural equation model showing object movement, self-movement, physical well-being, and social integration with the covariate age.

**Table 1 ijerph-19-14537-t001:** Descriptions of the test items (see Herrmann, 2018 (p. 15) and 2020 (p. 8–9) [1,32]). Note: 0 = no attempt completed, 1 = task completed once, 2 = task completed twice.

	MOBAK-KG	MOBAK-1-2
**Object movement**		
Throwing	The child throws six juggling balls at a target of 1.1 m height from a distance of 1.5 m with overhead throws.	The child throws six juggling balls at a target of 1.3 m height from a distance of 2.0 m.
Catching	The tester drops a small basketball to the ground from a height of 1.5 m so that the ball bounces back up at least 1.1 m from the ground. The child catches the ball after it has reached the highest point.	The tester drops a small ball to the ground from a height of 2.0 m so that the ball bounces back up at least 1.3 m from the ground. The child catches the ball after it has reached the highest point.
Bouncing	The child bounces a small volleyball with both hands and catches it again without losing the ball.	The child bounces a small basketball through a marked corridor (5.0 × 1.0 m) without losing the ball.
Dribbling	The child dribbles a futsal ball through a marked corridor (2.8 × 9.0 m) around two obstacles without stopping or losing the ball.	The child dribbles a futsal ball through a marked corridor (5.0 × 1.0 m) without losing the ball.
**Self-movement**		
Balancing	The child walks across an overturned long bench without stepping off the bench.	The child walks across an overturned see-sawing long bench without stepping off the bench.
Rolling	The child performs a forward roll down an inclined mat and is able to land fluently in a standing position on his/her feet.	The child performs a forward roll on a mat and is able to land fluently in a standing position on his/her feet.
Jumping	The child jumps a distance of 3.0 m on one foot, turns around, and jumps back 3.0 m on the other foot.	The child jumps between and beneath carpet tiles fluently with one leg between the tiles and with straddled legs beneath the tiles.
Running	The child runs forward along a corridor (0.6 m × 4.0 m) to a wall, touches it with his/her hand, and then runs back backwards.	The child moves sideways from one cone to another placed at a distance of 3 m from each other.

**Table 2 ijerph-19-14537-t002:** Descriptive analyses of sum scores of the motor competency domains, social integration, general HRQoL, and physical well-being.

	Preschool					Primary School				
	Overall		Boys	Girls		Overall		Boys	Girls	
	M [95% CI]	ICC	M [95% CI]	M [95% CI]	d	M [95% CI]	ICC	M [95% CI]	M [95% CI]	d
Object movement ^a^	4.0 [3.8; 4.1]	0.02	4.4 [4.2; 4.5]	3.5 [3.4; 3.7]	0.42	5.5 [5.4; 5.6]	0.12	5.9 [5.7; 6.0]	5.1 [5.0; 5.3]	0.43
Self-movement ^a^	4.5 [4.4; 4.7]	0.05	4.3 [4.1; 4.5]	4.8 [4.6; 5.0]	0.21	4.9 [4.8; 5.1]	0.14	4.8 [4.6; 5.0]	5.1 [4.9; 5.3]	0.14
Social integration ^a^	13.5 [13.4; 13.7]	0.19	13.4 [13.2; 13.5]	13.7 [13.6; 13.9]	0.17	13.7 [13.6; 13.8]	0.25	13.5 [13.3; 13.8]	13.9 [13.7; 14.1]	0.15
General HRQoL sum score ^b^	41.5 [41.2; 41.7]	0.04	41.4 [41.1; 41.8]	41.5 [41.1; 41.8]	0.004	41.2 [40.9; 41.5]	0.03	40.8 [40.4; 41.2]	41.5 [41.2; 41.9]	0.18
General HRQoL *t*-value ^1,b^	51.9 [51.4; 52.4]	0.02	51.8 [51.1; 52.5]	51.9 [51.2; 52.7]	0.02	51.7 [51.1; 52.2]	0.02	51.1 [50.3; 51.8]	52.3 [51.5; 53.1]	0.15
Physical well-being sum score ^c^	21.9 [21.7; 22.2]	0.04	22.3 [21.9; 22.6]	21.6 [21.1; 22.0]	0.27					
Physical well-being *t*-value ^1,c^	53.2 [52.6; 53.8]	0.003	54.1 [53.3; 54.8]	52.2 [51.3; 53.1]	0.33					

Note: M = mean, 95% CI = 95% confidence interval. Point ranges: object movement (0–8), self-movement (0–8), social integration (5–20), KIDSCREEN-10 sum score (10–50), KIDSCREEN physical well-being (5–25). ^1^ The sum score (range: 10–50) was transformed into t-values (mean: 50, standard deviation: 10). Higher values indicate better general health-related quality of life or physical well-being [37]. ^a^ Complete sample (preschool: N = 1163, primary school N = 880), ^b^ subsample 1 (preschool: N = 943, primary school N = 880), ^c^ subsample 2 (preschool: N = 384).

**Table 3 ijerph-19-14537-t003:** Data fit of the calculated models.

Model	Analysis	Sample	*n*	χ^2^	*df*	*p*	CFI	RMSEA
1a	MIMIC	SS 1 preschool	943	115.947	69	<0.001	0.961	0.027
1b	MIMIC	SS 1 primary school	880	92.865	69	0.029	0.967	0.020
2a	MGM	SS 1 preschool	943	201.069	160	0.015	0.971	0.023
2b	MGM	SS 1 primary school	880	180.123	160	0.132	0.977	0.017
3	MIMIC	SS 2 preschool	384	88.459	69	0.057	0.962	0.027
4	MGM	SS 2 preschool	384	175.645	160	0.188	0.971	0.023

Note: CFI = comparative fit index; RMSEA = root mean square error of approximation; MIMIC = structural equation model with covariate age; MGM = multigroup model; SS = subsample.

**Table 4 ijerph-19-14537-t004:** Intercorrelations between variables in model 1 (subsample 1).

Factors	Preschool (Model 1a)				Primary School (Model 1b)			
	(1)	(2)	(3)	(4)	(1)	(2)	(3)	(4)
(1) Object movement								
(2) Self-movement	0.74 ***				0.56 ***			
(3) Social integration	0.27 ***	0.26 ***			0.22 ***	0.18 **		
(4) General HRQoL	0.01	0.04	0.04		0.003	0.14 **	0.13 ***	
(5) Age	0.59 ***	0.47 ***	0.15 ***	−0.01	0.49 ***	0.33 ***	0.10	<0.01

Note: (1) object movement, (2) self-movement, (3) social integration, (4) general HRQoL. ** *p* < 0.01, *** *p* < 0.001.

**Table 5 ijerph-19-14537-t005:** Intercorrelations between the variables in model 2 (subsample 1).

Factors	Preschool (Model 2a)					Primary School (Model 2b)				
	(1)	(2)	(3)	(4)	(5)	(1)	(2)	(3)	(4)	(5)
(1) Object movement		0.85 ***	0.34 ***	−0.01	0.53 ***		0.64 ***	0.21 **	0.11	0.44 ***
(2) Self-movement	0.84 ***		0.22 *	0.01	0.44 ***	0.68 ***		0.12	0.12	0.26 ***
(3) Social integration	0.26 ***	0.27 ***		0.07	0.15 ***	0.32 ***	0.24 **		0.16 **	0.09
(4) General HRQoL	0.06	0.05	0.01		−0.02	−0.03	0.18 **	0.10		−0.001
(5) Age	0.71 ***	0.52 ***	0.15 ***	−0.02		0.58 ***	0.44 ***	0.10	−0.001	

Note: (1) object movement, (2) self-movement, (3) social integration, (4) general HRQoL. * *p* < 0.05, ** *p* < 0.01, *** *p* < 0.001. Girls below the diagonal, boys above the diagonal

**Table 6 ijerph-19-14537-t006:** Intercorrelations between the variables in model 3 (subsample 2).

Factors	(1)	(2)	(3)	(4)
(1) Object movement				
(2) Self-movement	0.79 ***			
(3) Social integration	0.29 ***	0.40 ***		
(4) Physical well-being	0.20 **	0.29 ***	0.07	
(5) Age	0.54 ***	0.39 ***	0.17 **	−0.04

Note: (1) object movement, (2) self-movement, (3) social integration, (4) physical well-being. ** *p* < 0.01, *** *p* < 0.001.

**Table 7 ijerph-19-14537-t007:** Intercorrelations between the variables in model 4 (subsample 2).

Factors	Boys				Girls			
	(1)	(2)	(3)	(4)	(1)	(2)	(3)	(4)
(1) Object movement								
(2) Self-movement	0.83 ***				0.88 ***			
(3) Social integration	0.40 ***	0.26 **			0.24 **	0.48 ***		
(4) Physical well-being	0.21 **	0.35 ***	0.05		0.15	0.34 **	0.18	
(5) Age	0.45 ***	0.29 ***	0.19 ***	−0.05	0.68 ***	0.58 ***	0.20 ***	−0.04

(1) Object movement, (2) self-movement, (3) social integration, (4) physical well-being. ** *p* < 0.01, *** *p* < 0.001.

## Data Availability

The data presented in this study are available on request from the corresponding author. The data are not publicly available due to the ethical guidelines of the Cantonal School Authorities.

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
