# Peer review of "Basic Motor Competencies of (Pre)School Children: The Role of Social Integration and Health-Related Quality of Life"

_ijerph, 2022, doi:10.3390/ijerph192114537_

Round 1

Reviewer 1 Report

General comments:

I would like to commend the authors on well-written and clearly presented manuscript. The research question/topic begins to fill an increasingly important research gap on the relationship between motor competence, children’s social development, and overall quality of life. I appreciate the holistic approach the author’s take.

There are a few comments below pertaining to specific line items, but in terms of more general feedback, I feel the discussion could be strengthened through support from existing literature and discussion of practical implications. For example, in lines 347-371, the discussion is more of a reiteration of the results as opposed to an explanation of ‘why.’ In lines 391-408, there is an attempt at explanation of the ‘why’, however, this should be supported by literature. After reading the discussion, I was left without a clear sense of how to apply your findings in a practical sense. I recommend including more content similar to lines 431-435. I also feel including an acknowledgement or discussion around how COVID-19 restrictions may have ultimately impacted your social integration results may be pertinent here.

Line/Page #

Comment

35

To suggest that BMC ‘guarantee’ basic capacity is somewhat bold. Perhaps a better word could be ‘support’ or ‘facilitate’? This statement suggests that is we have BMC, we will 100% engage in sports, which we know is just not true considering all we know about the role of motivation, confidence, etc.

47-49

Please include citation.

57-59

Please include citation.

63

After comma after ‘especially’.

97-98

There needs to be a bit more of a rationale above for why you have selected a typically developing population. After reading lines 72-81, I was expecting your sample to include non-typically developing participants.

141

Italicize ‘n’s

All tables

The tables are not numbered correctly. There are multiple table 2’s, 3’s, and 4’s. Further, with the exception of Table 1, the formatting does not look correct. 

241 & 247

An extra ‘L’ at the end of ‘Model’ 3 and 4

291

Error code?

304

Remove extra period before ‘and’.

318

Error code?

373

Remove comma after ‘due to the fact’.

374

Further to my comment above, you suggest here that motor competencies ‘may be important’, but above in line 35 you have guaranteed they result in sport participation.

395

Remove ‘since’.

Reviewer 2 Report

General comment

Through this article (“Basic motor competencies of (pre)school children: The role of social integration and health-related quality of life”) an interesting topic concerning aspects of children development/life, is highlighted. However, despite its merit, there are several flaws that diminish its overall quality. Specifically, my main concerns are (a) the data analysis, which is both incomplete (preliminary and/or other additional analyses are missing) and incoherently presented, (b) the writing style, which is rather confusing to the reader, since it is not as concise and comprehensive as it could/should be, (c) the way information is presented throughout the text (certain pieces of information are either unnecessary and/or wrongly placed, not to mention that there are too many repetitions of information), (d) the inconsistent usage of terms, e.g. basic motor competencies or motor competencies or motor competence or gross motor competencies and social integration or social relationships or social interactions, (e) the lack of citation(s) after important statements (additionally, some citations are wrongly placed in a sentence), (f) the selection of words/phrases/expressions is somewhat “weird” (the authors should have their manuscript proofread by an English native speaker).

Please refer to my specific comments below.

Accordingly, the article cannot be published in its current form. On the contrary, major revisions should be made to improve its quality and enable its publication.

Specific comments

Introduction

The main “problem” of the introduction is the fact that the authors fail to feature the originality of their research, mainly because they do not present a comprehensive literature review and also, they do not manage to “navigate” the reader through the multiple information presented in the introduction. In addition, it is not eloquently explained in this part why children of the two selected age groups (pre-school and primary school-aged children) present potential differences and therefore they must be examined separately. Similarly, gender differences are not thoroughly addressed, not in respect of literature concerning all three variables of interest.

Line 34. “MOBAK in German: Motorische Basiskompetenzen”

This reference is unnecessary here. This is an instrument measuring BMC.

Line 35 “[1]”

Can this citation support this statement?

Lines 37-38. “BMC are also the prerequisite for acquisition of sport-specific skills and positively influence a physically active lifestyle 38 over the lifespan [2].”

This citation is too old. There are more recent studies to be referred here.

Lines 41-42. “Therefore, a certain level of BMC is necessary for participation in social interactions in sport situations [1].”

This cannot be conducted by what it was claimed previously in this paragraph. In addition, I’m not sure that this statement can be supported by this citation.

Lines 43-49. “Mandatory (pre)school in particular … with new tasks and challenges.”

Information about the Educational system in Switzerland is unnecessary here. Besides, it is presented elsewhere.

In this paragraph, I believe that the authors would like to make the connection between BMC and schooling, since BMC are related to learning outcomes in Physical Education. Unfortunately, this is not evident. What is more, there is reference to social development, before addressing the second variable of interest, i.e., “Social interactions/integration”.

Line 52. “gross motor competencies [7].”

This term “sounds” weird. The authors refer to gross motor skills. Moreover, the citation used is too old. Please refer to a more recent one.

Lines 57-58. “While boys more often engage in physically active games, girls desire friendly closeness and cohesion more than boys”

Please add a citation to support this claim.

Lines 66. “(basic) motor competencies”

Please use the abbreviation presented before in the introduction, i.e., BMC. Please, use the abbreviations you initiate throughout the manuscript.

Lines 66-67. “The development of … children’s mental and physical health”

One sentence cannot constitute a paragraph (at least two sentences are needed). The information of this paragraph can be included previously in the introduction, specifically in the presentation of BMC and “social relationships” variables, respectively.

Lines 69-70. “Health-related quality of life (HRQoL) is a multi-dimensional and complex construct, described as individual’s perception of their position in life.”

Please add a citation.

Line 77. “health-related quality of life”

Use the abbreviation HRQoL which was initiated at the beginning of this paragraph.

Lines 80-81. “HRQoL is important to measure the subjective state of health of the child.”

A citation is required here (throughout the manuscript, the authors ought to make proper use of the citations by placing them in the correct part of the sentences).

Lines 87-90. “Low motor competencies can lead … and mental health.”

In this paragraph (which is again a one-sentence paragraph), the authors refer to motor competencies (and not BMC) and then at the end of the sentence refer to motor competence. There is an inconsistency here. Furthermore, mediation results are not clear. Please rephrase for more clarity.

Line 94-96. “with a focus on motor competencies and physical activity, show that children with 94 higher motor competencies and children who spent more time in moderate physical activity have better HRQoL levels [16,33].”

The reference of physical activity is quite unexpected here. It is weird to refer to a new contract at this part of the introduction, right before addressing the purpose of the study. School and physical activity settings are relevant to BMC development; however, this is not addressed properly in the introduction.

Materials and Methods

This section needs to be improved to enhance the clarity of the information presented. Additionally, the titles of certain subsections need to be reconsidered.

Lines 102-122. “In spring and summer 2021 … children assented to participate.”

This part includes information that refers either to the assessment “procedure” or to “participants” of the study.

Please, make sure that information is not repeated in the text, e.g., as it is the case for the information about the parents.

Line 124. “2.1 Sample”

“Participants” is better than “sample”.

Lines 128-129. “Inclusion criteria were the presence 128 of data of the BMC test and parent questionnaire.”

Did the parents give their consent for the assessment of BMC, or is it compulsory for PE?

Lines 138-141. “In Switzerland, … in primary school.”

I do not believe this information is useful. Please delete it.

Lines 174-175. “The internal consistency of the scale is satisfactory, with Cronbach’s alpha of .82 for preschool and .83 for primary school.”

Please add the respective citation.

Line 180. “in a subsample of N=943 preschool”

Why not in the total sample? Please provide an explanation.

Lines 185-186. “The internal consistency of the KIDSCREEN-10-instrument is acceptable with a Cronbach’s alpha of .73 for preschool and .76 for primary school (overall .74).”

Please add the citation(s) for these findings.

Lines 203-204. “Model 1 and Model 2 were calculated separately for preschool (a) and primary school (b). Model 3”

What kind of models? Please explain in detail (you can also present the models in figures).

Lines 207-209. “Reliability was examined for the scales of social integration (PIQ; Venetz et al., 2019), general HRQoL (KIDSCREEN-10 [43]) and physical well-being (KIDSCREEN-27 [19]) with Cronbach’s alpha above .7 considered acceptable”.

This information is repeated. It shouldn’t be in this subsection. It already appears above in the presentation of measures.

Lines 212-215. “The factorial validity of the MOBAK … in a study by 214 Herrmann et al. [12].”.

Again, this information should be in the parts where BMC and social integration assessments are described.

Lines 216-219. “Descriptive analyses of sum scores … physical well-being T-value”.

The title of the Table is too long. The scores (ranges) should appear at the note, bellow the Table.

Lines 237-240 & 249-252. “All parameters were estimated … was the same for boys 239 and girls.” & “All parameters were estimated … was the same for boys 239 and girls.”

Repetition of information. Please, find a way not to repeat this piece of information.

Line 251.”latent factors”

Is BMC the only latent factor in the analyses? A latent factor consisting of only two variables? Please explain more about the variables used in SEM analyses. Figures would be more informative than text information.

Lines 261-265. “Table 1: Descriptive analyses … well-being T-264 value.”

Why is the Table description here? Please delete it. Probably the authors would like to add in the text a phrase like “descriptive statistics will be calculated for all variables”. However, the title of such Table will be Table 2 not Table 1. Please check the Tables’ number throughout the text.

Results

Presentation of results is difficult to follow. It is not clear what analyses have been conducted. Information presented in the Tables is repeated, to some extent, in the text, e.g., correlation values. It must be rewritten in a more comprehensive and scientific way. Additional analyses could have been conducted, e.g., gender and age differences.

Discussion

There is no need to repeat the findings without discussing or explaining them (both the expected and the unexpected findings). Please add plausible explanations to interpret the results, e.g., why there are differences between genders in quality of life in primary school (of course in relation to differences in the other variables examined). If they are indeed gender differences, what does this mean for teachers who deal with children of both genders?

Please delete the statistical values from discussion. 

When refer to a study, also refer to its findings, e.g., “So far, differences in HRQoL between age and gender have only been studied from 8 years onwards [53], (line 358-359.).” & “…the relationship between motor competence and HRQoL in typically developing children has only been investigated in one study by Redondo-Tébar [16], (line 360-361)”.

The discussion does not actually feature the (practical) importance of the study’s key finding. Please enrich this part. Why the significant associations found in this study are important? What should we do (e.g., as teachers, practitioners) to take advantage of this finding?

Conclusion

It should be rewritten to feature the findings of the study and its practical implications.

Round 2

Reviewer 2 Report

General comment

I would like to congratulate the authors for the extensive revision of their manuscript. It has been greatly improved. However, there are a few parts of it which need to be further improved before publication. For that, the authors may find it useful to consider my specific comments below.

Materials and Methods

Line 102. “2.1 Participants”

(a)    What was the sampling method that was used? I believe that the sample in this study was a convenient sample. The authors must include this information.

(b)    Did you contact children outside your target age-group?

I’m asking this because of the 1334 preschoolers’ parents who gave their written consent and sent back the questionnaires, 1163 preschoolers were finally included in the analysis. Why? What about the rest? Were they outside the range of 55 to 80 months or this happened because of something else not mentioned here. Please be more specific. 

Line 124. “are measured via four items”

There are four items for each component (as seen in the Table). Please add “each” here for more clarity.

Line 135. “trained tester”

The word “examiner” is more suitable here.

Line 172. “(see [1,32])”

There is no need to include reference here. The reference was made in the text before.

In addition, I believe that the text in Table 1 would look better if it was aligned at the left (as we look at it).

Line 176. “…., t-tests and …”

Please clarify a bit more about the calculations of the t-tests (between which means?). Are the results of the t-tests included in the Results section?

Lines 188-189. “for both MOBAK-KG (Model 1a) and MOBAK-1-2 (Model 1b)”

It would be more informative to explain that this model was (separately) examined for both the age groups of interest.

Lines 186-189. “We used structural equation models to examine the relationships between the two MOBAK factors self-movement and object movement, social integration, and general HRQoL, with age as a covariate for both MOBAK-KG (Model 1a) and MOBAK-1-2 (Model 1b).”

It is better to start this part (Model 1) with a sentence like the above and then present the type of each variable.

Lines 190-192. “we summed up general HRQoL, transformed it into the t-value, and included it as a manifest variable in the model”

This is a repetition of information (Lines 182-184: “we summed up general HRQoL, transformed it into the t-value, and included it as a manifest variable in the model”). Please rephrase.

Lines 201-204. “In a subsample of N = 384 preschool children (subsample 2), we used the physical well-being subscale of the KIDSCREEN-27 instrument (5 items, [36]) to assess the children's physical well-being from the parents' perspective. The sum score of the 5 items was t-transformed into a manifest variable.”

Don’t we already know this?  Try to be as concise as you can.

Lines 225-228. “Due to the different instruments for the assessment of BMC for preschool and primary school, Model 1 and Model 2 were calculated separately for preschool (a) and primary school (b). Model 3 was calculated exclusively for preschool, as the subscale was used in a subsample of N = 384 preschool children only.”

This information should appear at the description of each model. It is repeated here.

Results

The information in this part (along with data analysis part) is not as specific as it should be. Please explain in more detail (and scientific style) all the analyses that were conducted and their results.

Line 245. “Table 3. Model fits of the calculated models”

Please correct “Model fits” to “model fit”.

Lines 250-253. “The structural equation model with object movement, self-movement, and social integration as latent constructs and general HRQoL (t-value of KIDSCREEN-10 sum score) as a manifest variable achieved a good model fit for both preschool (Model 1a) and primary school (Model 1b) (Table 3).”

This sentence can be shortened to a smaller sentence such as this: “both Model 1a (preschool) and Model 1b (primary school) fitted well to the data (Table 3)” or a similar one. All other information is unnecessary since you have already described the models in data analysis (do accordingly in other similar occasions). 

Line 254. “Table 4. Intercorrelations between the latent factors, the manifest factor, and with age as a covariate (Model 1, Subsample 1)”.

The title of Table 4 should be more concise, e.g., “Intercorrelations between variables in Model 1” (apply this in other Tables, too). 

Discussion

Line 324. “to have psychological issues, such as poor social skills or a decreased quality of life”.

Not “such as”. It is more accurate to write something like “…that may result from poor social skills or decreased quality of life”.

Line 326. “Redondo-Tébar and colleagues”

Please add the date of publication in parentheses.

Lines 329-331. “In contrast to previous studies in which motor competence instruments were used in a clinical context (e.g., MABC-2 [24]), our study used curricularly valid instruments that examine BMC in self-movement and object movement.”

This sentence is not a paragraph of its own. It should be positioned elsewhere.

Lines 343-344 & 357-359. “It is possible that general HRQoL in preschool is more strongly influenced by other factors, such as family factors (e.g., parents, siblings).” & “It is possible that general HRQoL in preschool is more strongly influenced by family factors (e.g., parents, siblings).’

These two sentences are written to support the same thing in different parts of the discussion. I understand that the authors want to explain findings regarding quality of life and well-being. I think that the explanation about the second one is lacking. Besides there is no comprehensive differentiation between the two constructs.

Lines 360-362. “A strength of the study is that the investigated constructs BMC (motor competence test; [1,32]), social integration (teacher perspective; [15]), and general HRQoL or physical well-being (parent perspective; [36,50]) were examined from different perspectives”.

No need to explain the perspectives again in parentheses. Besides these are explained in the sentence which follows.

Lines 369-371. “Since some dimensions of HRQoL are possibly more or less associated with motor competence or social integration of children, this could be a reason for the low correlations”.

This sentence does not make sense to me.

Lines 383-384. “Both BMC and social integration appear to be related to general HRQoL and physical well-being.”

Why is this sentence here? The limitation regarding casualty has been explained.

Line 385. “This also raises the pedagogical-didactic question of how to design a careful”

This part refers nicely to the practical implications of the study. The beginning of this paragraph should point out that the findings of the study raise ….
